# Insights into the Bacterial Diversity and Detection of Opportunistic Pathogens in Mexican Chili Powder

**DOI:** 10.3390/microorganisms10081677

**Published:** 2022-08-20

**Authors:** Yoali Fernanda Hernández Gómez, Jacqueline González Espinosa, Miguel Ángel Ramos López, Jackeline Lizzeta Arvizu Gómez, Carlos Saldaña, José Alberto Rodríguez Morales, María Carlota García Gutiérrez, Victor Pérez Moreno, Erika Álvarez Hidalgo, Jorge Nuñez Ramírez, George H. Jones, José Luis Hernández Flores, Juan Campos Guillén

**Affiliations:** 1Facultad de Ciencias Naturales, Universidad Autónoma de Querétaro, Av. De las Ciencias s/n, Santiago de Querétaro 76220, Mexico; 2Facultad de Química, Universidad Autónoma de Querétaro, Cerro de las Campanas s/n, Santiago de Querétaro 76010, Mexico; 3Secretaría de Investigación y Posgrado, Centro Nayarita de Innovación y Transferencia de Tecnología (CENITT), Universidad Autónoma de Nayarit, Tepic 63173, Mexico; 4Facultad de Ingeniería, Universidad Autónoma de Querétaro, Cerro de las Campanas s/n, Santiago de Querétaro 76010, Mexico; 5Department of Biology, Emory University, Atlanta, GA 30322, USA; 6Centro de Investigación y de Estudios Avanzados del IPN, Irapuato 36824, Mexico

**Keywords:** bacteria diversity, β-lactamases, Mexican chili powder, *Bacillaceae*, *Enterobacteriaceae*

## Abstract

Chili powder is the most frequently consumed spice in Mexican diets. Thus, the dissemination of microorganisms associated with chili powder derived from *Capsicum annuum* L. is significant during microbial quality analysis, with special attention on detection of potential pathogens. The results presented here describe the initial characterization of bacterial community structure in commercial chili powder samples. Our results demonstrate that, within the domain Bacteria, the most abundant family was *Bacillaceae*, with a relative abundance of 99% in 71.4% of chili powder samples, while 28.6% of samples showed an average relative abundance of 60% for the *Enterobacteriaceae* family. Bacterial load for aerobic mesophilic bacteria (AMB) ranged from 10^4^ to 10^6^ cfu/g, while for sporulated mesophilic bacteria (SMB), the count ranged from 10^2^ to 10^5^ cfu/g. *Bacillus cereus sensu lato* (*s.l.*) was observed at ca. ˂600 cfu/g, while the count for *Enterobacteriaceae* ranged from 10^3^ to 10^6^ cfu/g, *Escherichia coli* and *Salmonella* were not detected. Fungal and yeast counts ranged from 10^2^ to 10^5^ cfu/g. Further analysis of the opportunistic pathogens isolated, such as *B. cereus s.l.* and *Kosakonia cowanii,* using antibiotic-resistance profiles and toxinogenic characteristics, revealed the presence of extended-spectrum β-lactamases (ESBLs) and Metallo-β-lactamases (MBLs) in these organisms. These results extend our knowledge of bacterial diversity and the presence of opportunistic pathogens associated with Mexican chili powder and highlight the potential health risks posed by its use through the spread of antibiotic-resistance and the production of various toxins. Our findings may be useful in developing procedures for microbial control during chili powder production.

## 1. Introduction

Chili powder is a condiment obtained from pulverized air-dried fruit produced by plants of the *Solanaceae* family, in particular from chili pepper (*Capsicum annuum* L.). This pepper is cultivated around the world in phenotypically diverse varieties with specific and attractive traits. The peppers are used to produce various chili powder formulations of commercial interest, and they have also found utility as components of a variety of foodstuffs, as food colorants, and in the pharmaceutical industry [1,2,3,4,5,6]. As chili powder is a ready-to-eat condiment, there are issues related to its quality and the safety of its use which are in turn related to the geographical areas from which the chili powder is obtained. Agricultural and processing practices involved in chili powder production have been studied [7,8,9,10,11,12,13,14,15,16], and these studies indicate the need for a comprehensive assessment of possibilities for contamination of chili powders by diverse microorganisms, especially those with pathogenic traits that may cause food-borne illnesses in the consumers.

Many significant environmental and ecological factors can affect chili powder quality. Among such factors are the diversity of microorganisms associated with various chili powders, so the community structure of those microbial populations and their potential pathogenicity could be analyzed by 16S rRNA Next-Generation Sequencing [17]. However, systematic research on microbial communities in chili powder has been limited. Studies using conventional approaches based on microbial culture-dependent methods have revealed that chili powder could be a habitat for an enormous variety of microorganisms [7,8,9,10,11,12,13,14,15,16]. Thus, it has been reported that chili powder can contain up to 10^7^ CFU/g of aerobic mesophilic bacteria [10,18], and that aerobic spore-forming bacteria can reach 10^5^–10^7^ CFU/g [12,15,18]. *Bacillus*
*cereus* was reported to be ˂10^4^ cfu/g [13,15,18,19]. Similar studies have been conducted for the detection of pathogenic members of the *Enterobacteriaceae* family in paprika powder [10,11,13,16,18]. 

The emergence of antimicrobial resistance is a very important health challenge and a significant public health concern. In this regard, diverse *B. cereus sensu lato (s.l.)* strains with toxinogenic (cereulide, cytotoxin K, hemolysin BL and nonhemolytic enterotoxin) and multidrug-resistant characteristics have been detected as potential pathogenic bacteria in chili powder, paprika, and other spices of different geographical origins [13,14,15,19]. Furthermore, *B. cereus s.l*. strains are naturally resistant to β-lactam antibiotics because of their content of β-lactamases, such as extended-spectrum β-lactamases (ESBLs) and Metallo-β-lactamases (MBLs) [20,21,22,23,24,25]. Moreover, due to their importance as potential pathogens and as carriers of antibiotic resistance determinants, members of the family *Enterobacteriaceae* have been monitored as microbiological indicators of the effectiveness of hygienic conditions employed during chili powder production [10,11,13,16,18]. ESBLs and carbapenemases associated with these microbes play an important role during clinical infections [26,27,28]. It is notable in this regard that an outbreak of human salmonellosis was associated with paprika powdered potato chips in Germany [16]. 

The aim of the present study was to use standard and molecular microbial procedures to analyze Mexican chili powder samples elaborated from *Capsicum annuum* L. We examined the bacterial and fungal load, and the bacterial community structure using 16S rDNA high-throughput sequencing, as well as antibiotic-susceptibility profiles. We also tested for the presence of β-lactamases and the toxinogenic potential for isolated opportunistic pathogens. Our results indicate that the characterization of bacterial community structure is a robust and reliable microbiological diagnostic tool whose implementation can be used to assess the quality of hygienic procedures employed in the production of chili powder and to reduce the dissemination of opportunistic pathogens during chili powder production and commercialization.

## 2. Materials and Methods

### 2.1. Bacterial and Fungal Load

We analyzed seven samples of commercial chili powder elaborated from *Capsicum annuum* L. obtained from different suppliers. Samples A, B, and C are a red chili powder obtained from a local market in Queretaro city, México. Sample D is a red chili powder obtained from a producer in Zacatecas state, México. Samples E, F, and G are a red-hot chili powder obtained from local producers at Queretaro state, México. For each sample, 500 g of chili powder were collected in sterile Erlenmeyer flasks and kept at 10 °C until analysis. The content of aerobic mesophilic bacteria (AMB), sporulated mesophilic bacteria (SMB), *Enterobacteriaceae* (E), and fungi and yeasts (F/Y) from each sample was determined according to Mexican regulations (NOM-092, NOM-113, and NOM-111) and microbiological Guidelines for Food according to the FDA’s *Bacteriological Analytical Manual* [29]. For further analysis, 10 g of each sample was weighed, and 90 mL of peptone diluent (0.1%) were added in an Erlenmeyer flask and samples were homogenized during 5 min. This suspension was used to obtain decimal dilutions and the bacterial and fungal colonies on duplicate plates of appropriate medium (see below) were counted after 24–48 h of incubation. Data were recorded as CFU/g of chili powder.

### 2.2. Salmonella and E. coli Detection

*Salmonella* was investigated according to the FDA’s *Bacteriological Analytical Manual* [29]. For the pre-enrichment, 25 g of chili powder sample were added to 225 mL of lactose broth, blended for 10 min, and incubated at 35 °C for 22 h. After pre-enrichment, 1 mL of sample was added to 9 mL of Rappaport Vassialidis broth (BD Difco) and tetrathionate broth (BD Difco) and incubated at 43 °C for 24 h. After enrichment, the samples were streaked on bismuth sulfite agar and xylose lysine desoxycholate agar, and incubated at 37 °C for 24 h. Colonies of interest were tested on triple sugar iron (TSI) and lysine iron (LIA) agar and incubated at 37 °C for 24 h. Colonies exhibiting typical phenotypes on TSI and LIA were characterized by assays of urease, oxidase, phenylalanine decarboxylase, Voges-Proskauer, indole, and citrate. *E. coli* was investigated according to the FDA’s *Bacteriological Analytical Manual* using the most probable number (MPN) method [29]. For these studies, 25 g of chili powder sample were added to 225 mL of lactose broth and blended for 10 min. This suspension was used to obtain decimal dilutions in lactose broth and incubated at 35 °C for 24–48 h. Samples with presumptive positive (gas) tubes were inoculated in lauryl tryptose MUG broth and incubated at 35 °C for 24–48 h for confirmation, and the most probable number (MPN) was calculated.

### 2.3. B. cereus s.l. Detection

Detection and isolation of *B. cereus s.l*. was performed according to the FDA’s *Bacteriological Analytical Manual* [29]. In these analyses, 10 g of chili powder sample were blended in 90 mL of peptone (0.1% *w*/*v*; Difco Laboratories; Detroit, MI, USA) and incubated at 80 °C for 10 min. Decimal dilutions were then inoculated onto duplicated agar plates containing *B. cereus* agar base (Sigma-Aldrich, México) supplemented with 100 mL/L of egg yolk emulsion and 10 µg/mL of polymyxin B and incubated at 35 °C for 24 h. Data were recorded as CFU/g of chili powder. Suspected *B. cereus s.l*. colonies with typical phenotypes were selected and confirmed in the same culture medium. *B. cereus* ATCC 10876 was used as reference strain for phenotypic tests and phylogenetic analysis was performed for representative isolates by the previous methods reported by our research group [19,30].

### 2.4. Isolation and Characterization of Ampicillin-Resistant Bacteria

To select ampicillin-resistant colonies of *B. cereus s.l*. and members of *Enterobacteriaceae*, the culture media described above were supplemented at a concentration of 100 µg/mL of ampicillin. Plates were incubated at 37 °C for 24 h and colonies were isolated. Representative isolates putatively identified as members of the family *Enterobacteriaceae* were identified by PCR using primers 27F and 1492R for 16s rRNA [31]. 

The antibiotic susceptibilities of 50 colonies for each bacterial group were tested using the disc diffusion method determined by the criteria of CLSI standards [32]. *B. cereus* ATCC 10876 and *Escherichia coli* XL1blue were used as controls. Antibiotic discs (Oxoid, México) containing amikacin (amk 30 μg), ampicillin (amp 10 μg), amoxicillin/clavulanic acid (amc 20/10 µg), carbenicillin (cab 100 µg), cefalotin (cft 30 µg), cefotaxime (ctx 30 μg), chloramphenicol (chl 30 μg), ciprofloxacin (cip 5 μg), clindamycin (cdm 30 µg), dicloxacillin (dcx 1 µg), erythromycin (ery 15 μg), gentamicin (gen 10 μg), imipenem (ipm 10 µg), meropenem (mem 10 µg), netilmicin (net 30 µg), nitrofurantoin (ntf 300 µg), norfloxacin (nfx 10 µg), penicillin (pen10 U), tetracycline (tet 30 μg), trimethoprim-sulfamethoxazole (sxt 25 μg), and vancomycin (vcm 30 µg) were used for susceptibility testing. The antibiotic activity was expressed as the mean of inhibition diameters (mm) produced in duplicate experiments and resistance profiles are shown as percentage of the total number of isolated colonies that showed resistance to a particular antibiotic.

For toxin production by *B. cereus s.l.* colonies, the Duopath^®^ Cereus Enterotoxins test (Merck, Kenilworth, NJ, USA) was used to detect the nonhemolytic enterotoxin (Nhe) and hemolysin BL (HBL) according to the supplier’s specifications [33]. All strains were confirmed by PCR for the presence of representative enterotoxin genes (*nheA* and *hblC*) following the methodology reported previously [34]. Detection of extended-spectrum β-lactamases (ESBLs) belonging to class A and Metallo-β-lactamases (MBLs) belonging to class B were performed by PCR using Phusion high-fidelity DNA polymerase (Thermo Scientific; Waltham, MA, USA) following the supplier’s recommendations. The primers for the class A and B *bla* genes of *B. cereus* species were synthesized as; BLA1F 5′-TGCTAAAAATAGGAATATGCGTTGG-3′ and BLA1R 5′-CCTTAACTATAACTTYMGTTGCC-3′, for class A. For class B as BLA2F 5′-GGRKTATGTGTWRCTTTACTAGG-3′ and BLA2R 5′- CATTCATTTACRTAMGCATCCG-3′ and were predicted to yield products of 894 bp and 629 bp, respectively. PCR conditions were as follows: after a hot start (2 min, 95 °C), 35 cycles at 95 °C for 30 s, 55 °C for 30 s, and 72 °C for 1 min, final elongation step at 72 °C for 5 min. The primers used for *bla* genes of isolated colonies for *Enterobacteriaceae* were bla-SHV.SE/AS; TEM-164.SE and TEM-165.AS; universal CTX-M-U1/U2 reported previously [35]. PCR products were analyzed by electrophoresis in 2.0% agarose (Sigma-Aldrich; St. Louis, MO, USA) and visualized on a UV transilluminator. Amplicons were sequenced at Macrogen Inc. (Seoul, Republic of Korea). Gene sequences were analyzed using MEGA X using the neighbor-joining method and the evolutionary distances were computed using the Jukes–Cantor method, and compared with sequences representative of ESBLs and MBLs using the BLDB database [36,37,38,39,40].

### 2.5. Bacterial Community Analysis

Using a culture-dependent approach, duplicated 10 g portions of each chili powder samples were added to 90 mL of peptone diluent (0.1%) and samples were homogenized during 5 min. Decimal dilutions were then inoculated onto four solid culture media; nutrient agar (Difco Laboratories; Detroit, MI, USA), Mueller Hinton agar (Bioxon, México), tryptic soy agar (Difco Laboratories; Detroit, MI, USA), and brain heart infusion agar (Bioxon). All bacterial colonies obtained for each culture medium were processed for genomic DNA (gDNA) purification using ZymoBIOMICS^TM^ DNA Miniprep Kit (Zymo Research, Irvine, CA, USA), and in the last step, genomic DNA for each culture medium was pooled together and duplicated DNA samples for each chili powder sample were obtained. The gDNA samples were processed and analyzed with the ZymoBIOMICS^®^ Targeted Sequencing Service (Zymo Research, Irvine, CA, USA). Bacterial 16S ribosomal RNA gene targeted sequencing was performed using the Quick-16S™ NGS Library Prep Kit (Zymo Research, Irvine, CA, USA) and sequenced on Illumina^®^ MiSeq™ with a v3 reagent kit (600 cycles). For the bioinformatic analysis unique amplicon sequence variants were inferred from raw reads using the Dada2 pipeline [41]. Potential sequencing errors and chimeric sequences were also removed with the Dada2 pipeline. Taxonomy assignment was performed using Uclust from Qiime v.1.9.1 [42,43] with the Zymo Research Database. *Accession number*. DNA sequences were deposited in NCBI as BioSample accession PRJNA847909.

## 3. Results

### 3.1. Bacterial Community Analysis

As a first step to obtain insights into bacterial diversity in chili powder, we used a culture-dependent approach followed by 16S rRNA gene analysis. Sequencing of duplicate samples as described in Materials and Methods yielded an average of 49,546 to 66,091 high-quality reads representing 14 operational taxonomic units (OTUs) for bacterial genera at a 97% level of similarity. Members of the *Bacillaceae* were found in all chili powder samples, and in 71.42% of samples, the genus *Bacillus* was the most abundant, with an average relative abundance of 99 to 100% (Figure 1). However, 28.6% of samples (D and F samples) showed an increased relative abundance for members of the *Enterobacterales*, with average values of 42.5% for sample D and ca. 77% for sample F. In samples D and F, the *Bacillaceae* showed a relative abundance of 52% and 23%, respectively (Figure 1). 

Within the family *Enterobacteriaceae* for sample D, the average relative abundances for representative genera were as follows: *Citrobacter* (6.5%); *Cronobacter* (1.5%); *Enterobacter* (5%); *Escherichia/Shigella* (4.5%); *Klebsiella* (0.65%); *Kluyvera* (0.84%); *Kosakonia* (3%); Leclercia (0.9%); *Pluralibacter* (1%); and *Siccibacter* (4%). For the *Erwiniaceae* family, *Erwinia* and *Pantoea* showed average relative abundances of 0.9% and 12.5%, respectively. For the *Pseudomonadaceae* family, *Pseudomonas* was detected at 0.85% relative abundance. For the *Enterobacteriaceae* in sample F, the genus with the highest relative abundance was *Enterobacter* with an average of 76%, followed by *Klebsiella* at 0.60%. Our results clearly demonstrate that using 16S rDNA sequencing to obtain bacterial diversity is a reliable microbiological diagnostic tool which can be employed during chili powder production.

### 3.2. Bacterial and Fungal Load

As an additional approach and to corroborate the relative abundance results for Bacillales and Enterobacterales, we analyzed the bacterial load following the guidelines of The International Commission on Microbiological Specifications for Foods (ICMSF) [44].

The content of aerobic mesophilic bacteria (AMB), sporulated mesophilic bacteria (SMB), *Enterobacteriaceae* (E), fungi and yeasts (F/Y), *Salmonella* and *E. coli* from seven chili powder samples was determined as described in Materials and Methods. As shown in Table 1, AMB counts for the chili powder samples ranged from 6.3 × 10^4^ to 1.2 × 10^6^ CFU/g. Samples with the highest levels of AMB contamination were D and E, followed by samples A, B, C, and G. The sample with the lowest AMB count was sample F. In accordance with the microbiological criteria for The International Commission on Microbiological Specifications for Foods (ICMSF) [44], all the chili powder samples have marginal quality (10^4^–10^6^ CFU/g), where values above of 10^6^ CFU/g indicate unacceptable and values of ≤10^4^ CFU/g indicate acceptable quality.

As an indicator of hygienic quality and in concordance with the 16S rRNA gene analysis, the results for *Enterobacteriaceae* showed that samples D and F exceeded the microbiological criteria for ICMSF (≤10^4^ CFU/g), while the rest of the samples with values of ≤10^3^ CFU/g showed acceptable quality for this group of microorganisms (Table 1). Our analysis demonstrated undetectable levels of *E. coli* and *Salmonella*. The results for fungi and yeasts (F/Y) showed significant levels of contamination in samples C, D, E, and F that exceeded the microbiological criteria for ICMSF (≤10^4^ CFU/g); only samples A, B, and G showed values of ≤10^4^ CFU/g with acceptable quality.

In general, sporulated mesophilic bacteria (SMB) are part of the microbiota resident in soil and are not considered as one of the microbiological quality criteria for spices. Nevertheless, in our study and in concordance with 16S rRNA gene analysis, this bacterial group was considered for further analysis. The results show that SMB counts ranged from 1.0 × 10^2^ to 3.5 × 10^5^ CFU/g; the sample with the highest level of this bacterial group was sample D (Table 1).

### 3.3. Isolation and Characterization of Opportunistic Pathogens

We next focused our efforts on the detection and isolation of members of the sporulated mesophilic bacteria (SMB) and members of *Enterobacteriaceae*, as these microbial groups might contain species that act as opportunistic pathogens. Thus, we first measured the levels of *Bacillus cereus sensu lato* (*s.l*) in all chili powder samples, utilizing the phenotypical characteristics mentioned in Materials and Methods. Except for sample D, with an average abundance of 600 CFU/g, the abundances in the other samples were at the limit of detection (LOD), viz. 100 CFU/g. From these results, presumptive *B. cereus s.l*. colonies from all chili powder samples were selected based on their ampicillin resistance phenotype. For the taxonomic confirmation of these ampicillin-resistant *B. cereus s.l.* colonies, we used the tRNA^Cys^-PCR strategy previously published [19,30]. The results of this analysis were used to construct the neighbor-joining phylogenetic tree shown in Figure 2A, in comparison with other relevant members of the genus *Bacillus*. Figure 2A shows that several isolates selected from the samples were phylogenetically related to *Bacillus thuringiensis* (Bc02, Bc03, Bc04, Bc05, Bco7), while Bc01 appeared to be most closely related to *Bacillus cereus*. Isolates Bc06, Bc08, Bc09, and Bc10 formed a clade which shared a node with a clade containing *B. thuringiensis* (Figure 2A). 

A similar analysis was performed to characterize members of the family *Enterobacteriaceae*. Based on ampicillin resistance and taxonomic analysis, we identified a number of isolates as members of the species *Kosakonia cowanii*. The 16S rRNA similarity was above 99% for these isolates (designated as P23–27F to P56–27F in Figure 2B). 

The presence of enterotoxin genes in the isolates of *B. cereus s.l.* was investigated by PCR detection of *hblC* and *nheA* as representative genes. The PCR results showed that the isolates were positive for both toxin genes for at least 95% of the isolates. Molecular results were confirmed by the Duopath test for detection of toxin production and a positive result for the Hbl toxin was observed in 90% of the isolates, while the Nhe toxin was detected in 100% of the isolates, including the strains in which the PCR results were negative for enterotoxin genes.

Dissemination of antibiotic resistance via opportunistic pathogens through the consumption of foods is a threat to public health, thus we analyzed the antibiotic profile of the *B. cereus s.l.* and *K. cowanii* isolates from the chili powder samples to evaluate the potential health risk these isolates might pose. Results of this analysis are shown in Figure 3 and confirm the following results based on interpretation of CLSI breakpoints for the antibiotics tested.

*B. cereus s.l.* isolates showed a high percentage of resistance to the β-lactam antibiotics tested, such as penicillin G (pen), ampicillin (amp), carbenicillin (cab), cefalotin (cft), cefotaxime (ctx), dicloxacillin (dcx), and amoxicillin/clavulanic acid (amc) at 100% of the isolates. Additionally, these isolates were significantly resistant to trimethoprim-sulfamethoxazole (sxt) at 100%, tetracycline (tet) at 90%, erythromycin (ery) at 77%, clindamycin (cdm) at 74%, chloramphenicol (chl) at 42%, and at lower percentages, amikacin (amk) at 6%, gentamicin (gen) at 6%, netilmicin (net) at 3%, and nitrofurantoin (ntf) at 3%. On the other hand, *B. cereus s.l.* isolates were susceptible to ciprofloxacin (cip) at 100%, norfloxacin (nfx) at 100%, vancomycin (vcm) at 100%, and Imipenem (ipm) and meropenem (mem) at 100%. Figure 3 shows the resistance for each of the antibiotics tested as a color-coded bar graph.

In contrast to the foregoing results, *K. cowanii* isolates showed a lower percentage of resistance for the β-lactam antibiotics tested, in that penicillin G (pen), ampicillin (amp), and carbenicillin (cab) were observed at 100%, while for cefalotin (cft), cefotaxime (ctx), dicloxacillin (dcx), amoxicillin/clavulanic acid (amc), imipenem (ipm), and meropenem (mem) the isolates were susceptible. Additionally, erythromycin (ery), clindamycin (cdm), and vancomycin (vcm) at 100% resistance were observed. The *K. cowanii* isolates showed marked susceptibility to the remaining antibiotics (Figure 3). Based on the β-lactam antibiotics resistance results in both bacterial groups, we decided to investigate the presence of β-lactamase enzymes. 

Detection of extended-spectrum β-lactamases (ESBLs) belonging to class A and Metallo-β-lactamases (MBLs) belonging to class B was performed by PCR as described in Materials and Methods. The PCR results indicated that for *B. cereus s.l.* isolates, both β-lactamase classes A and B were present in 35.5% of isolates, while β-lactamase class A alone was present in 58.1%, and 6.45% of the isolates showed the presence of β-lactamase class B only. Phylogenetic relationships for the β-lactamases are shown in Figure 4A for representative isolates of *B. cereus s.l*. in comparison with β-lactamase sequences obtained from the database. In the case of the *K. cowanii* isolates, the PCR test for β-lactamases detected TEM β-lactamases (class A) in 100% of the isolates. The phylogenetic relationships between the β-lactamase sequences from the representative isolates in comparison with sequences from the genome database are shown in Figure 4B.

## 4. Discussion

The results presented here indicate clearly that chili powders obtained from different geographical regions, which presumably involve different ecological factors, and agricultural practices, make it necessary to monitor the commercial products for possible contamination with opportunistic microbial pathogens. Our studies identified potential pathogens with varying antibiotic-resistance and toxigenic phenotypes, and these properties could pose health risks for humans and for particular animal populations. Therefore, the assessment of microbial content must be a priority in certifying chili powder production for local commercialization, import, and export.

Specifically, our results, based on 16S rRNA analyses and confirmed by microbial load analyses, show that, in accordance with ICMSF, all of the chili powder samples examined were of marginal quality (10^4^–10^6^ CFU/g) in terms of AMB contamination. Moreover, 28.6% of samples exceeded the microbiological criteria (≤10^4^ CFU/g) for *Enterobacteriaceae*, while 57% of samples exceeded the microbiological criteria (≤10^4^ CFU/g) for fungi and yeasts. A high percentage of the AMB present in the chili powder samples examined here were members of the family *Bacillaceae*. Similar results have been reported in paprika spice, in samples from different countries, where the predominant bacterial species identified by 16S rRNA were members of the *Bacillaceae* family [12]. Although fungi and yeasts were not identified in our study, the presence of fungi and yeast has been confirmed, some with the capability of producing mycotoxins, in paprika [10]. Therefore, compressive deep studies are necessary to understand the microbial community structure and metabolic function for the establishment of microbiological criteria in chili powder and if they pose a public health concern.

The question arises, what might be the sources of microbial contamination of commercial chili powder preparations. Several possibilities exist. First, a potential source of microbial contamination in chili powder could be principally from the soil and water used during agricultural practices [45]. Second, microbial contamination could also result from the use of inadequate sanitary procedures in the preparation of commercial chili powder. Fecal matter associated with animals that might come into contact with the chili plants might also produce microbial contaminants. The potential pathogenicity of these contaminants might well be enhanced by the documented and increasing incidence of the release of antibiotics from different sources into water and soil, thus increasing the concentration of antibiotic-resistant organisms in these environments [45,46]. Therefore, dissemination of antibiotic resistance in opportunistic pathogens through the production of chili powder elaborated from *Capsicum annuum* L. poses a potential threat to public health, and that threat clearly deserves additional investigation. 

As indicated above, *B. cereus sensu lato* (*s.l.*) has been shown to be a contaminant of a number of different spices produced in different geographical areas [13,14,15,19]. On the one hand, in our studies, chili powder sample D, with the highest SMB count (3.5 × 10^5^ CFU/g, Table 1), showed relatively low levels (˂600 cfu/g) of *B. cereus s.l.* Data provided by the European Food Safety Authority (EFSA) specify that food-borne outbreaks caused by *B. cereus* have been associated with bacterial concentrations above 10^5^ cfu/g of foodstuff [47]. However, based on our results and compared with other studies [13,14,15,19], the *B. cereus* strains detected in chili powder are highly toxigenic, exhibiting a prevalence of Hbl and Nhe toxins in 90% and 100% of isolates, respectively. Moreover, the strains exhibited multidrug-resistance in almost all β-lactams antibiotics tested, as well as 100% of the *B. cereus* strains being resistant to trimethoprim-sulfamethoxazole, 90% to tetracycline, 77% to erythromycin, 74% to clindamycin, and 42% to chloramphenicol. Although our results show that *B. cereus s.l.* load on chili powder could not be a health risk factor for the consumers, its toxin production capabilities reflected that a monitoring program is necessary to avoid potential bacterial outbreaks on chili powder. 

Diverse mechanisms of selection for antimicrobial resistance on opportunistic pathogens have been studied in at least two directions; antibiotic-inactivating enzymes and nonenzymatic mechanisms and the evidence show a threat to public health during hospital-acquired infection [26,27,28]. Therefore, the high prevalence of members of the *Enterobacteriaceae* family and *B. cereus* producing extended-spectrum β-lactamases (ESBLs) and Metallo-β-lactamases (MBLs) make new demands for β-lactamases typing. So, the results obtained here reflect and confirm previous reports [48] about the presence of β-lactamases class A in *B. cereus* with the ability to hydrolyze penicillins and inhibited by clavulanic acid, while class B inactive cephalosporins but is not inhibited by clavulanate, therefore a combination of both β-lactamases in *B. cereus* appear to be primarily responsible for β-lactams resistance profiles, except for carbapenems.

All of the *K. cowanii* isolates showed resistance to erythromycin, clindamycin, and vancomycin. The presence of class A TEM β-lactamases was also detected in our isolates, consistent with other reports [49,50], which showed resistance of these species to some penicillins, but not to dicloxacillin, cephalosporins, amoxicillin/clavulanic acid, Imipenem, or meropenem. The genus *Kosakonia* arose from a reclassification of *Enterobacter* genus [49,50,51,52], and diverse species (*K. radicincitans, K. sacchari, K. oryzae, K. cowanii, K. arachidis*) have been characterized as plant growth-promoting bacteria [51]. Some species have also been characterized as phytopathogens causing diverse plant diseases [51]. Of particular relevance to the present study is the indication that some instances of human infection have been associated with *K. radicincitans* and *K. cowanii* [49,50,52]. It was suggested that these infections arose from external sources. Our results demonstrate that *K. cowanii* isolated from chili powder could be a relevant microorganism with specific genetic traits; however, additional efforts are necessary to evaluate its capability in this context. 

On the other side, members of the *Bacillaceae* family, such as the *B. subtilis* group, have emerged with special interest for the ability to produce diverse bioactive metabolites with antimicrobial activities, food preservatives, plant growth-promoting activities, bioremediation, therapeutics agents, and biopesticides [53]. In the same way, members of *B. cereus s.l*, such as *B. thuringiensis*, have diverse phenotypes relevant as biological agent during formulation of bioinsecticides and also novel valuable bioproducts [54]. Therefore, chili powder could be a habitat for diverse bacteria with these metabolic abilities mentioned before and must be explored.

In conclusion, the results obtained in this study show that characterization of bacterial diversity, through diverse molecular and microbiological methods, in chili powder is relevant to consider for microbiological criteria, although with ecological factors during plant cultivation it is necessary to understand agricultural practices to avoid contamination of pathogenic microorganisms with antibiotic-resistance and toxinogenic capabilities that could be a factor risk for the consumers or environmental spread during commercialization. Further studies into the taxonomic relationship of isolates obtained in this work will no doubt be provided by comparative genomics to understand its metabolic and pathogenic capabilities. 

## Figures and Tables

**Figure 1 microorganisms-10-01677-f001:**
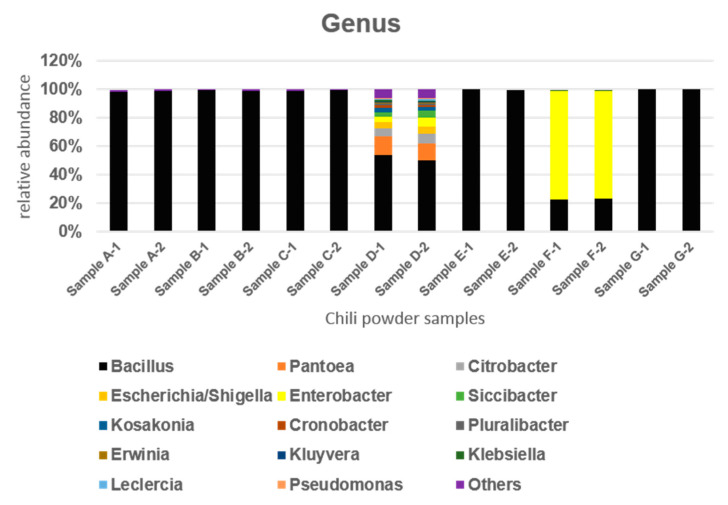
Taxonomic composition in seven commercial chili powder samples. The bar graph represents the relative abundance in percentage and depicts the average community composition of bacteria in the samples. The 16S rRNA genes were amplified and sequenced in duplicate as described in Materials and Methods.

**Figure 2 microorganisms-10-01677-f002:**
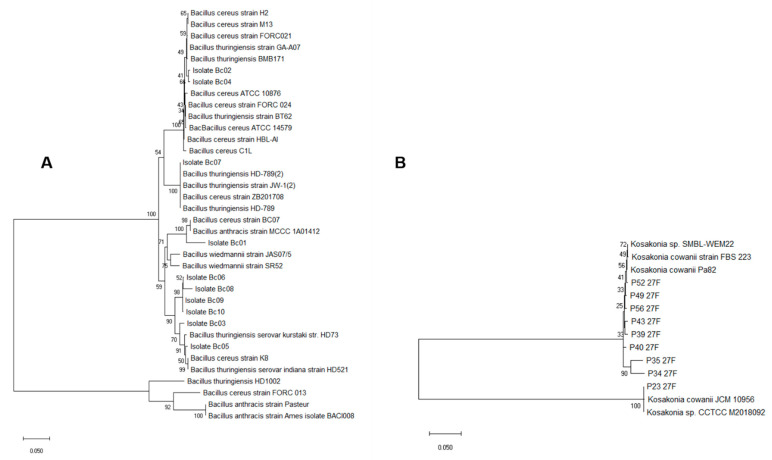
Phylogenetic analysis for *B. cereus s.l*. (**A**) and *K. cowanii* (**B**) isolates. Analyses were obtained in MEGA X, using the neighbor-joining method and the evolutionary distances were computed using the Jukes–Cantor method [40].

**Figure 3 microorganisms-10-01677-f003:**
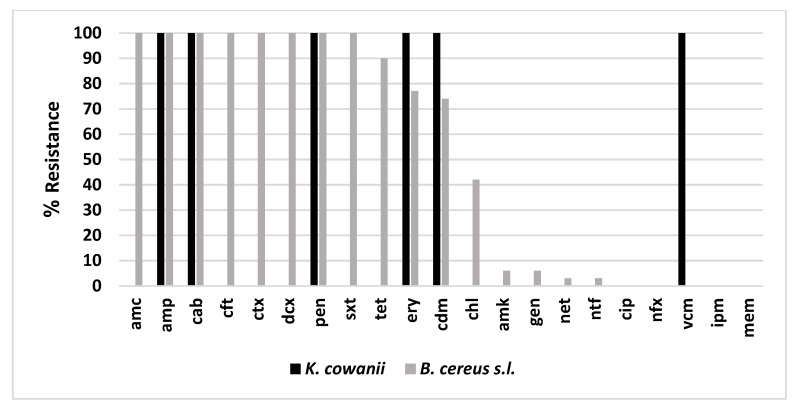
Antibiotic resistance profile. The bar graph represents the percentage of the isolates of each bacterial species that were resistant to a particular antibiotic tested.

**Figure 4 microorganisms-10-01677-f004:**
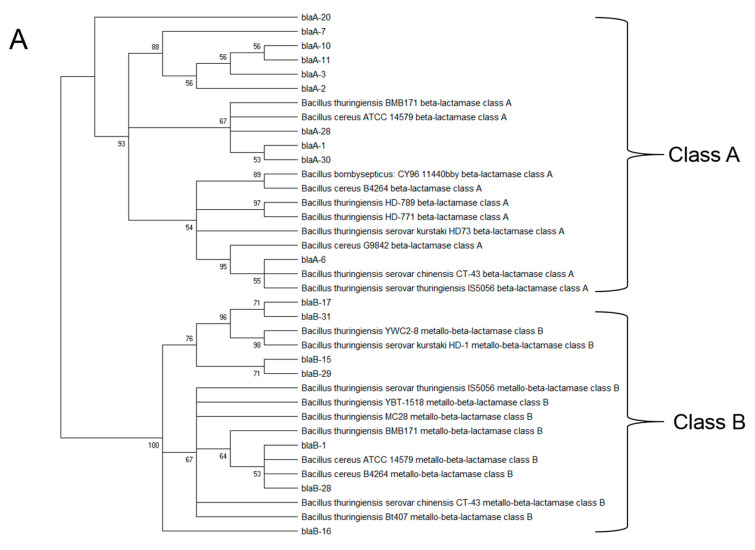
Detection of extended-spectrum β-lactamases (ESBLs) belonging to class A and Metallo-β-lactamases (MBLs) belonging to class B in *B. cereus s.l.* (**A**) and *K. cowanii* isolates (**B**) from chili powder. β-lactamases from isolates are indicated as *blaA*/*blaB* and isolate number (**A**), and as blaTEM in (**B**). Class A beta-lactamase SHV was included for the phylogenetic analysis.

**Table 1 microorganisms-10-01677-t001:** Bacterial and fungal load in chili powder samples.

Sample	AMB (CFU/g)	SMB (CFU/g)	F/Y (CFU/g)	*Enterobacteriaceae* (CFU/g)	*E. coli*	*Salmonella*
A	2.0 × 10^5^	1.5 × 10^3^	2.7 × 10^2^/4.2 × 10^2^	1.1 × 10^3^	ND *	ND *
B	1.6 × 10^5^	7.5 × 10^2^	1.8 × 10^3^/9.2 × 10^2^	4.2 × 10^2^	ND	ND
C	1.14 × 10^5^	2.4 × 10^3^	1.7 × 10^5^/4.3 × 10^4^	3.7 × 10^3^	ND	ND
D	1.2 × 10^6^	3.5 × 10^5^	5.9 × 10^4^/4.0 × 10^4^	1.2 × 10^5^	ND	ND
E	1.0 × 10^6^	4.3 × 10^3^	4.3 × 10^5^/5.5 × 10^4^	4.8 × 10^3^	ND	ND
F	6.3 × 10^4^	1.0 × 10^2^	1.1 × 10^5^/3.5 × 10^4^	1.4 × 10^4^	ND	ND
G	1.2 × 10^5^	4.5 × 10^2^	2.8 × 10^3^/7.5 × 10^2^	9.4 × 10^2^	ND	ND

* ND, not detected.

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
