# Peer review of "Insights into the Bacterial Diversity and Detection of Opportunistic Pathogens in Mexican Chili Powder"

_microorganisms, 2022, doi:10.3390/microorganisms10081677_

Round 1

Reviewer 1 Report

Title : may be better because the obtained and the discussed results are focused on B cereus.    - In the abstract  the meaning of abbreviations must be included.   - Reference 17 is not about the lack of chilli studies. Thus , other references must be checked.   - The humidity of sampled chilli may help to understand better the results , although the fungi were detected but not discussed, but justified the title of the article.   -The pathogenesis potencial of each cited and detected microbe should  be less inflated, although antibiotic resistant genes were detected an compared.   - The findinding of enterobacteriacea is important as a fecal contamination and also associated with the harvesting and storage of studiend chilli samples.   -  Are there some addictive compounds added to the final product ? they vary ?   -The growth promoting  kosakonia and the use of B. thurigienis in agriculture.must be somehow discussed to human health and their detections.   The article is well written, and presents good methodology.   -Although the article presents Mexican data it may add the knowledge of low literature data and may be published with  modifications mentioned above.

Author Response

Thank you for allowing us to revise our manuscript entitled  “Insights into the bacterial diversity and detection of opportunistic pathogens in Mexican chili powder ” and for useful comments.

Reviewer 1

1.- Title : may be better because the obtained and the discussed results are focused on B cereus.

Our answer:

Thank you for your comments. Our study was focus on bacterial diversity, these results were interesting to provide bacterial and fungal load, and additionally opportunistic pathogens isolated were characterized. So that, we think that the title correspond to these results.    

2.- In the abstract  the meaning of abbreviations must be included. 

Our answer:

Thank you for your comments. Your suggestion was followed.

 3.- Reference 17 is not about the lack of chilli studies. Thus , other references must be checked.   –

Our answer:

Thank you for your comments. Your suggestion was followed. These lines were extended to “Many significant environmental and ecological factors can affect chili powder quality. Among such factors are the diversity of microorganisms associated with various chili powders, so that the community structure of those microbial populations and their potential pathogenicity could be analyzed by 16S rRNA Next-Generation Sequencing [17]”.

4.- The humidity of sampled chilli may help to understand better the results.

Our answer:

Thank you for your comments. Moisture content in original chili samples were not evaluated because initially we focused on analysis of bacterial diversity for diverse methods, including direct DNA extraction from the chili samples, so that stored condition for the chili samples could be change with the time. However, we agree with your observation where moisture content could be critical for microbial growth, although resistance mechanism at low moisture content has been detected in members of Enterobacteriacea family (Lehmacher, A., J. Bockemühl, and S. Aleksic, Nationwide outbreak of human salmonellosis in Germany due to contaminated paprika and paprika-powdered potato chips. Epidemiology and Infection, 2009. 115(3): p. 501-511).

5.-although the fungi were detected but not discussed, but justified the title of the article.

Our answer:

Thank you for your comments. Although we detected the presence of fungi and yeasts in our study, were not identified since it was not our original goal. However, and due to a previous a report that some fungi are capable to produce mycotoxins in paprika [10], a comprehensive study is necessary to understand these fungal community structure and if they pose a health concern.

In discussion, some  lines are included: Although fungi and yeasts were not identified in our study, has been confirmed the presence of fungi and yeast, some with the capability of producing mycotoxins in paprika [10]. Therefore, compressive deep studies are necessary to understand the microbial community structure and metabolic function for the establishment of microbiological criteria in chili powder and if they pose a public health concern.    

6.-The pathogenesis potencial of each cited and detected microbe should  be less inflated, although antibiotic resistant genes were detected and compared.  

Our answer:

Thank you for your comments. However, we consider that our results show significant finding for chilli powder consumers and the inherent health risk occasioned by the presence of these potential pathogens. However, some text was changed.

7.- The findinding of enterobacteriacea is important as a fecal contamination and also associated with the harvesting and storage of studiend chilli samples.   -  Are there some addictive compounds added to the final product ? they vary ?

Our answer:

Thank you for your comments. The chili samples analyzed did not contain additional addictive compounds. This is an important point because microbial diversity could be change in elaborated mix of chili samples. However, in following studies we want to include diverse ecological variables to analyze changes in community structure of these microbial populations. 

8.-The growth promoting  kosakonia and the use of B. thurigienis in agriculture must be somehow discussed to human health and their detections.

Our answer:

Thank you for your comments.  In discussion, some  lines are included:

  1. Our results demonstrate that cowanii isolated from chili powder could be a relevant microorganism with specific genetic traits; however, additional efforts are necessary to evaluate its capability in this context.
  2. On the other side, members of Bacillaceae family, such as subtilis group have emerged with special interest for the ability to produce diverse bioactive metabolites with antimicrobial activities, food preservatives, plant growth-promoting activities, bioremediation, therapeutics agents and biopesticides [53]. In the same way, members of B. cereus s.l, such as B. thuringiensis have diverse phenotypes relevant as biological agent during formulation of bioinsecticides and also novel valuable bioproducts [54]. Therefore, chili powder could be a habitat for diverse bacteria with these metabolic abilities mentioned before and must to be explored.    

9.- The article is well written, and presents good methodology.   -Although the article presents Mexican data it may add the knowledge of low literature data and may be published with  modifications mentioned above.

Our answer:

Thank you for your comments.

Reviewer 2 Report

Paper investigated possible health risk related to toxin content and development of antibiotic-resistance in chili powder produced in Mexico.

Paper is well written and subject is very interesting. In addition, I guess this kind of research can be useful to suggest new methodologies improving control system in the field and safeguarding consumers.

I have no remarks to report. Work seems to me very clear and well structured in all parts.  

Author Response

Thank you for allowing us to revise our manuscript entitled  “Insights into the bacterial diversity and detection of opportunistic pathogens in Mexican chili powder ” and for useful comments.

Paper investigated possible health risk related to toxin content and development of antibiotic-resistance in chili powder produced in Mexico.

Paper is well written and subject is very interesting. In addition, I guess this kind of research can be useful to suggest new methodologies improving control system in the field and safeguarding consumers. I have no remarks to report. Work seems to me very clear and well structured in all parts. 

Our answer:

Thank you for your comments.

Reviewer 3 Report

A rather interesting manuscript submitted for review dealing with Mexican chili powder microbiota. Interesting results including 16S rRNA library sequencing and antibiotic-resistance profiles, toxinogenic characteristics and analysis of antibiotic resistance determinants are presented. However, the work needs to be improved. The first problem is justification for the need for such a study. You can past in this sentence: "dissemination of antibiotic resistance via opportunistic pathogens associated with **** is a significant public health concern" everything you want including cucumbers, books, hates, roses ect...  Besides, microbiological criteria for ICMSF (≤104 CFU/g) is important, but they probably refer to the end product. If you recalculate the load in the end product (consider the mass fraction of chili powder in the final dish) you will see that the danger may be exaggerated...

I think that there is another problem, the coverage of which could improve this work. There is a lot of publication (and this opinion is also common among ordinary people) about antimicrobial activity of chili powder, especially against enterobacteria. Among your samples there are two with extremely high proportion of this potential patogenes. Could you arrange a simple test (like disk test with antibiotic resistance) to compare activity of chili powder samples A, B, C, D, F against enterobacteria? The results of this test will be interesting in any case. I do not require such an experiment, it is just an option to improve the manuscript. Anyway, the problem of antimicrobial activity of chili powder shoud be presented more widely at least in Introduction and Discussion.

In general, the work is read with interest, especially due to the combination of molecular and classical microbiological approaches, especially those related to the analysis of the determinants of antibiotic resistance.  Below are a few notes.

201-202  More details needed about taxonomy assignment;

209-210  For communities with such low diversity, it is necessary to use ASV (amplicon sequence variant), not OTU 97%. Probaly it could be a good idea to present the general tree with all dominant representatives and indication of its presence in particular samples (see the fig below). It is just an option if you want.

 .......       It must be borne in mind that, according to strict criteria, taxonomy assignment by 16S rRNA gene is possible only up to the genus rank;

283-285 and in other places...  Calculate how many nucleotide positions  B.c. and B.t.  (especially those from the same cluster) ... Too few, I'm afraid. The resemblance to B.t. requires special discussion, since insect pathogens belong to them;

Fig. 2   and in other places...  Indicate please the type of distance you used;

361    I'm not sure that you have enough material for such generalizations;

381-382   Everything is everywhere (Bejerink).

Author Response

Thank you for allowing us to revise our manuscript entitled  “Insights into the bacterial diversity and detection of opportunistic pathogens in Mexican chili powder ” and for useful comments.

A rather interesting manuscript submitted for review dealing with Mexican chili powder microbiota. Interesting results including 16S rRNA library sequencing and antibiotic-resistance profiles, toxinogenic characteristics and analysis of antibiotic resistance determinants are presented. However, the work needs to be improved.

1.-  The first problem is justification for the need for such a study. You can past in this sentence: "dissemination of antibiotic resistance via opportunistic pathogens associated with **** is a significant public health concern" everything you want including cucumbers, books, hates, roses ect...  Besides, microbiological criteria for ICMSF (≤104 CFU/g) is important, but they probably refer to the end product. If you recalculate the load in the end product (consider the mass fraction of chili powder in the final dish) you will see that the danger may be exaggerated...

Our answer:

Thank you for your comments. The text in abstract was changed to “Chili powder is the most frequently consumed spice in Mexican diets. Thus, the dissemination of microorganisms associated with chili powder derived from Capsicum annuum L. is significant during microbial quality analysis, with special attention on detection of potential pathogens”.

Also, Our results indicate that the characterization of bacterial community structure is a robust and reliable microbiological diagnostic tool whose implementation can be used to assess the quality of hygienic procedures employed in the production of chili powder and to reduce the dissemination of opportunistic pathogens during chili powder production and commercialization.

Althought, there is not data on percapite consume for chili powder in Mexico and other countries, is relevant and frequently consumed in diverse foodstuffs, detection of potential pathogens is not exaggerated, we suggest  the following paper:  Lehmacher, A., J. Bockemühl, and S. Aleksic, Nationwide outbreak of human salmonellosis in Germany due to contaminated paprika and paprika-powdered potato chips. Epidemiology and Infection, 2009. 115(3): p. 501-511).

2.-I think that there is another problem, the coverage of which could improve this work. There is a lot of publication (and this opinion is also common among ordinary people) about antimicrobial activity of chili powder, especially against enterobacteria. Among your samples there are two with extremely high proportion of this potential patogenes. Could you arrange a simple test (like disk test with antibiotic resistance) to compare activity of chili powder samples A, B, C, D, F against enterobacteria? The results of this test will be interesting in any case. I do not require such an experiment, it is just an option to improve the manuscript. Anyway, the problem of antimicrobial activity of chili powder shoud be presented more widely at least in Introduction and Discussion.

Our answer:

Thank you for your comments.

Although antimicrobial activity of Capsicum genus is well known, this activity if often related to pure capsaicin or concentrated extracts (Dorantes, 2000; Gomes, 2019; Brito-Argáez, 2009), in our study we analyzed chili powder with a way less concentrated antimicrobial compounds and therefore it is not expected any considerable antimicrobial activity able to alter microbial quality of the samples, additionally, there is a previous report on a Salmonella outbreak due to paprika powder [16]. Furthermore, bacterial load in samples D and E strongly suggest that little or no antimicrobial activity in those samples.

Lidia Dorantes, Raul Colmenero, Humberto Hernandez, Lydia Mota, Maria Eugenia Jaramillo, Elizabeth Fernandez, Claudia Solano, Inhibition of growth of some foodborne pathogenic bacteria by Capsicum annum extracts, International Journal of Food Microbiology, Volume 57, Issues 1–2, 2000, Pages 125-128, ISSN 0168-1605, https://doi.org/10.1016/S0168-1605(00)00216-6.

Rafael Gomes Von Borowski, Karine Rigon Zimmer, Bianca Franco Leonardi, Danielle Silva Trentin, Rodrigo Campos Silva, Muriel Primon de Barros, Alexandre José Macedo, Simone Cristina Baggio Gnoatto, Grace Gosmann, Aline Rigon Zimmer, Red pepper Capsicum baccatum: source of antiadhesive and antibiofilm compounds against nosocomial bacteria, Industrial Crops and Products, Volume 127, 2019, Pages 148-157, ISSN 0926-6690, https://doi.org/10.1016/j.indcrop.2018.10.011.

Brito-Argáez L, Moguel-Salazar F, Zamudio F, González-Estrada T, Islas-Flores I (2009) Characterization of a Capsicum chinense Seed Peptide Fraction with Broad Antibacterial Activity. Asian Journal of Biochemistry 4: 77-87.

 However, the aim of the present study was to use standard and molecular microbial procedures to analyze Mexican chili powder samples elaborated from Capsicum annuum L. However, is a good idea to prove antimicrobial  activity for chili powder using diverse methodologies to obtain biomolecules with antimicrobial activities.  So that, a manuscript with these results could be better in these context.

3.- In general, the work is read with interest, especially due to the combination of molecular and classical microbiological approaches, especially those related to the analysis of the determinants of antibiotic resistance.  Below are a few notes.

Our answer:

Thank you for your comments.

4.- 201-202  More details needed about taxonomy assignment;

Our answer:

Thank you for your comments. Taxonomy assignment is based on Zymo Research Database pipelines, which is a commercial sequencing service. We do not have more detail about this curated database.

5.- 209-210  For communities with such low diversity, it is necessary to use ASV (amplicon sequence variant), not OTU 97%. Probaly it could be a good idea to present the general tree with all dominant representatives and indication of its presence in particular samples (see the fig below). It is just an option if you want.

 .......       It must be borne in mind that, according to strict criteria, taxonomy assignment by 16S rRNA gene is possible only up to the genus rank;

Our answer:

Thank you for your comments, is an excellent idea, however interpretation could be difficult. Actually, we have diverse isolated bacteria, and molecular markers are under evaluation for taxonomic profile.

6.- 283-285 and in other places...  Calculate how many nucleotide positions  B.c. and B.t.  (especially those from the same cluster) ... Too few, I'm afraid. The resemblance to B.t. requires special discussion, since insect pathogens belong to them;

Our answer:

Thanks you for your comments. The genomes of species belonging to the B. cereus group show very high similarity, so that the molecular methodology used in this work is to confirm the bacterial group and show some relationship between closest isolated, therefore is necessary use multilocus typing or genome sequencing with phenotyping traits for taxonomic interpretation. 

7.- Fig. 2   and in other places...  Indicate please the type of distance you used;

Our answer:

Thanks you for your comments.  Additional text was added at Figure 2. Phylogenetic analysis for B. cereus s.l (A) and K. cowanii (B) isolates. Analysis were obtained in MEGA X, using the Neighbor-Joining method and the evolutionary distances were computed using the Jukes-Cantor method [40].

8.-361    I'm not sure that you have enough material for such generalizations;

Our answer:

Thanks you for your comments. The results obtained are a minimum representative sample from a wide chili powder diversity in Mexico, thus requiring additional efforts to extend similar microbial analysis throughout the country, where chili powder is produced.

381-382   Everything is everywhere (Bejerink).

Our answer:

Thanks you for your comments.

Round 2

Reviewer 3 Report

OK, just a comment on your answer about antimicrobial activity of chili powder samples. You wrote "  bacterial load in samples D and E strongly suggest that little or no antimicrobial activity in those samples". The idea I meant was to compare antimicrobial activity of samples D and E with those, where enterobacterial load was low. But I am not insisting...